# Number of Teeth Is Related to Craniofacial Morphology in Humans

**DOI:** 10.3390/biology11040544

**Published:** 2022-04-01

**Authors:** Elias S. Oeschger, Georgios Kanavakis, Alina Cocos, Demetrios J. Halazonetis, Nikolaos Gkantidis

**Affiliations:** 1Department of Orthodontics and Dentofacial Orthopedics, University of Bern, CH-3010 Bern, Switzerland; elias.oeschger@zmk.unibe.ch; 2Department of Orthodontics and Pediatric Dentistry, UZB—University School of Dental Medicine, University of Basel, CH-4056 Basel, Switzerland; georgios.kanavakis@unibas.ch; 3Department of Orthodontics and Dentofacial Orthopedics, Tufts University School of Dental Medicine, Boston, MA 02111, USA; 4Department of Orthodontics, School of Dentistry, National and Kapodistrian University of Athens, GR-11527 Athens, Greece; cocos_alina@hotmail.com (A.C.); dhalaz@dent.uoa.gr (D.J.H.)

**Keywords:** growth/development, tooth agenesis, hypodontia, oligodontia, morphogenesis, morphometrics, tooth development

## Abstract

**Simple Summary:**

In modern humans, congenital absence of one or more permanent teeth has a prevalence of 22.6% when considering the third molars and of 6.4% when not. Its high prevalence, in conjunction with evolutionary findings pinpointing to a steady reduction in teeth number, raises the question whether the congenital absence of teeth in modern humans is an evolutionary trend rather than an anomaly. Previous studies have shown that modern humans with less teeth also have smaller faces; however, the association between teeth number and craniofacial morphology remains unclear. Here, we show that less teeth are associated with a flatter profile and a decreased facial height. These findings support the claim of a broader relationship between number of teeth and overall craniofacial development and have evolutionary implications, since face reduction comprises also an evolutionary trend in humans.

**Abstract:**

One of the most common dental anomalies in humans is the congenital absence of teeth, referred to as tooth agenesis. The association of tooth agenesis to craniofacial morphology has been previously investigated but remains unclear. We investigated this association by applying geometric morphometric methods in a large sample of modern humans. In line with previous studies, we report here that a reduced teeth number is linked to a less convex profile, as well as to a shorter face. The effects were similar for males and females; they increased as the severity of the tooth agenesis increased and remained unaltered by the inclusion of third molars and of allometry in the analysis. Furthermore, in cases with tooth agenesis only in the maxilla, there was no detectable effect in mandibular shape, whereas maxillary shape was affected independently of the location of missing teeth. The robustness of the present sample along with the shape analysis and the statistical approach applied, allowed for thorough testing of various contributing factors regarding the presence but also the magnitude of effects. The present findings suggest a relationship between number of teeth and overall craniofacial development and have evolutionary implications.

## 1. Introduction

The absence of formation of one or more permanent teeth during development, namely, tooth agenesis, has a common occurrence in humans. It has a prevalence of 6.4% and is shown to be more frequent in females than in males (1.22:1 females/males ratio). Geographical differences have also been detected, with African populations showing the highest risk (regional prevalence: 13.4%). Mandibular second premolars are the most commonly affected, followed by maxillary incisors and maxillary second premolars [1,2].

Genetics are considered the primary cause of tooth agenesis [3]. Similar genetic factors are also involved in other dental anomalies, such as tooth shape and size discrepancies, suggesting a shared genetic background among various aspects of dental development [4,5]. Specific genes are shown to be connected to more than 150 syndromes, in which tooth agenesis coexists with other features. However, tooth agenesis most commonly appears to be a sporadic isolated trait or tends to segregate in families [3,6]. Several genes involved in human tooth agenesis have also been shown to affect craniofacial bone morphogenesis [7,8].

A phenotypic relationship between tooth agenesis and craniofacial morphology has been reported in several recent studies [8,9,10,11,12]. However, there are also conflicting reports, both, regarding the presence of an effect, as well as its magnitude and direction [9,13,14]. These contrasting findings can be attributed to differences in sample composition, sample size, and applied methodology. Most studies, for example, have made their assessments applying conventional cephalometric methods, which have many shortcomings [15]. Among other limitations, conventional cephalometry uses only a small part of the information required to adequately describe complex shapes, such as craniofacial structures, by measuring selected angles and linear distances, defined by specific landmarks. There are several limitations to this approach including: (a) measurements are affected by the local anatomy of landmark areas, (b) the decision regarding the landmarks and measurements to be used and those that should impact decisions is subjective and might be biased, (c) often measurements for similar outcomes are contrasting, and (d) meaningful comparisons of actual craniofacial shape differences between subjects or time points are difficult to perform using a small number of angles or distances. A recent systematic review and meta-analysis summarized the evidence from conventional cephalometric studies and rated it as moderate to low. Moderate evidence supports the conclusion that individuals with tooth agenesis present a smaller SNA angle, but no solid conclusion could be drawn for any other outcome [16]. Since differences in craniofacial shape due to genetic linkages are likely to be subtle, a solid methodology is required for their detection. For this purpose, alternative approaches suitable for investigating small differences, such as geometric morphometrics, are proposed [11,12,17,18,19]. Two recent geometric morphometric studies assessing the effect of tooth agenesis on craniofacial size revealed a significant decrease in size in cases of general tooth agenesis [11] and of isolated third molar agenesis [12]. Furthermore, a recent study utilizing similar methods in a group of 100 individuals with tooth agenesis identified possible differences in the craniofacial shape between tooth agenesis and control individuals. However, as stated also by the authors of this study, the limited sample size creates a need for more research towards this direction [9].

Thus, the aim of the present study was to investigate the association between the number of permanent teeth and the shape of the craniofacial complex. To make this evaluation, geometric morphometric methods were applied in a large sample of modern humans. The analysis was performed twice, with and without considering the absence of third molars.

## 2. Materials and Methods

### 2.1. Ethical Approval

The Ethics Commission of the Canton of Bern, Switzerland (Project-ID: 2018-01340), and the Research Committee of the School of Dentistry, National and Kapodistrian University of Athens, Greece (Project-ID: 281, 9 February 2016), reviewed the protocol and granted their approval for this retrospective observational case-control study. The STROBE guidelines were followed for reporting. The participants whose data were used in the study had previously provided written informed consent.

### 2.2. Sample

The present study constitutes part of a larger project. In this context, the current sample was studied and analysed for the purpose of other relevant publications [2,11,12,20] and has been previously described in detail [11]. Therefore, only sample information necessary for the understanding of this manuscript is conveyed here. The selected population was obtained from archived consecutive orthodontic patient records between 2002 and December 2017, at the following orthodontic clinics: (a) University of Bern, Switzerland; (b) National and Kapodistrian University of Athens, Greece; (c) two private practices in Athens and two in Thessaloniki, Greece; and (d) one private practice in Biel, Switzerland. The following inclusion criteria were applied:Permanent tooth agenesis (congenitally missing teeth), without considering the third molars.No craniofacial malformations, syndromes, systemic diseases, or any other anomalies affecting craniofacial morphology, as reported in the subjects’ medical record.Individuals older than 8 years of age and younger than 40 years of age. For all individuals younger than 12 years old at the time of the pre-treatment radiograph, radiographs obtained at older ages were also examined [21,22].European (White) ancestry.Lateral cephalometric radiograph in maximal intercuspation of adequate clinical diagnostic quality and with a reference ruler at the mid-sagittal level.Panoramic radiographs of adequate diagnostic quality.No history of interventions known to influence craniofacial morphology, such as orthodontic treatment.Absence of any other severe dental anomaly regarding tooth number, size, or form in any tooth except for third molars.Individuals where the reason of absence of any tooth was definite.

The medical and dental history, the photographs, and the radiographs of each individual were reviewed. Relevant data were recorded in an Excel sheet (Microsoft Excel, Microsoft Corporation, Redmond, WA, USA). The patterns of permanent tooth agenesis were recorded using the TAC system [2,23].

As reported previously [11], out of more than 8000 orthodontic patient files reviewed, 404 individuals (238 females, 166 males; Median age 13.0 years, range: 8.0–38.3 years) with tooth agenesis of teeth other than third molars comprised the final study population. Thorough analyses of the patterns of tooth agenesis of a very similar group, excluding third molars [2], and those of third molars [20] have been published in previous reports. The current sample differs by less than 3% from the aforementioned study populations [11,20]. From the same archives, a control sample of 404 individuals without tooth agenesis (not considering third molars) was retrieved and matched to the agenesis group for age (within 6 months), sex, and geographic origin. A detailed agenesis distribution of the sample by age and sex is provided in Appendix A.

### 2.3. Shape Assessment

The frequency distribution of the number of missing teeth in the present sample (*n* = 808) has been published previously [11]. In brief, 493 individuals had tooth agenesis, including third molars (mean: 4.1 missing teeth per subject), whereas 315 individuals had no tooth agenesis. When disregarding third molars, 404 individuals had tooth agenesis (mean: 2.7 missing teeth per subject), and 404 did not.

As reported previously [11,12], the craniofacial areas of interest were described with landmark configurations identified on lateral cephalometric radiographs. Shape analysis of these configurations was performed on Viewbox 4 software (dHAL software, Kifissia, Greece) using geometric morphometric methods [19]. The shape of the entire craniofacial configuration (not including the posterior and superior part of the cranium) and the individual shapes of the cranial base, the maxilla, and the mandible comprised the primary outcomes tested in this study (Figure 1). The alveolar bone area was not included in the selected landmark configuration, since missing teeth may affect alveolar bone morphology [24]. Furthermore, to capture the anterior end of the maxillary and the mandibular structures towards the dentition and avoid local effects due to missing teeth, four fixed landmarks were placed along the overall level of the cementoenamel junction of teeth and the level of the overall alveolar bone margin. In total, 127 landmarks were used. Eleven were positioned at extreme structures of local anatomy such as the anterior (ANS) and posterior (PNS) nasal spines and were considered fixed landmarks. All other 116 landmarks were initially distributed equidistantly on fifteen curves, describing the shape outlines under investigation. These were defined as semilandmarks [25] and were, thus, allowed to slide from their initial position along their corresponding curve. Semi-landmarks do not have a biological interpretation and do thus not show homology between different samples. In order to achieve the largest possible homology between corresponding landmarks in all sample configurations, a sliding process of all semi-landmarks was performed. Sliding aimed to minimize bending energy and reduce shape variability, against a reference configuration representing the average shape of all configurations in the sample. During this iterative process, an average shape was computed and used as a reference for each next sliding circle [26]. In each circle, sliding was repeated six times and the entire process was repeated three times until no change in individual shapes was detectable.

Subsequently, all final landmark configurations were superimposed using partial generalized Procrustes superimposition, in order to transform landmark coordinates to shape coordinates [27]. The resulting coordinates, namely, Procrustes coordinates, described the location of each subject in shape space [28]. In order to reduce the amount of shape variables and allow for interpretable statistical analyses, the Procrustes coordinates were subjected to a Principal Component Analysis (PCA). This is a standard step in geometric morphometric investigations [28,29,30].

### 2.4. Statistical Analysis

The statistical analysis was conducted with IBM SPSS statistics for Windows (Version 28.0. Armonk, NY: IBM Corp) and Viewbox 4 software. A two-sided significance test was carried out at an alpha level of 0.05. A Bonferroni correction was applied on the level of statistical significance, were applicable.

Sexual dimorphism in craniofacial shape configurations was assessed with permutation tests (100,000 permutations without replacement) on Procrustes distances between male and female group means.

To test for the effect of size-related changes in the tested craniofacial configurations, known as allometry, the centroid size (cs) [31,32] of each craniofacial configuration was regressed against the shape Principal Components (PCs) describing the respective configuration (cranial base, maxilla, mandible, and entire facial configuration). The individual regression models were run separately for males and females. The number of PCs that were used to provide shape information for each configuration was determined with the broken stick method [33].

To assess the effect of missing teeth on the shape of each craniofacial configuration (cranial base, maxilla, mandible, and entire facial configuration), respective multivariate regression models were developed (general linear model and full factorial) with shape PCs as dependent variables and age and number of missing teeth as predictors. In presence of allometry, similar regression models were performed after removing the allometric component from the shape PCs of each craniofacial configuration. This was performed by using the standardized residuals of multivariate regression of shape PCs on centroid size to rerun the general lineal models described above for each respective configuration. Due to the presence of sexual dimorphism and in light of the developmental differences between males and females, analyses were performed separately for each sex. To visualize shape differences in relation to tooth agenesis, shape information (with and without allometry) provided by the selected shape PCs was used to create shape morphings representing cases with no or few missing teeth and cases with maximum number of missing teeth. To be able to visualize the effects without allometry, the unstandardized residuals of shape PCs were used because they carry the information for shape coordinates. The subsequent morphings were superimposed with Procrustes superimposition and are presented in figures.

The potential effect of the location of tooth agenesis on the shape of the corresponding maxillary and mandibular structures was tested in two subsamples. One subsample consisted of individuals without tooth agenesis or with agenesis only in the maxilla, and the other consisted of individuals without tooth agenesis or with agenesis only in the mandible. A multivariate regression model was built to test the effect of sex, age, and number of missing teeth, on the maxillary and mandibular shapes of subjects that either had no tooth agenesis or had tooth agenesis only in the maxilla. A similar model was used for subjects that either had no tooth agenesis or had tooth agenesis only in the mandible. By comparing the outcomes of these two models, we investigated potential local over generalized effects of tooth agenesis on the respective maxillary and mandibular structures. Due to sample size considerations, both sexes were tested together in the particular models.

All the analyses described above were performed twice, once accounting for missing third molars and once disregarding their presence.

### 2.5. Method Error

To assess the error of the method, the study methodology was repeated for 30 randomly selected samples, at least two weeks after the first digitization. The shape coordinates of these 30 subjects from the second digitization were compared to the coordinates from the first digitization, in two ways. At first, the mean Procrustes distance between the first and second shape configurations was determined through permutations tests (100,000 permutations) and was minimal, indicating no systematic error (*p* > 0.5). Moreover, random digitization error was assessed as the percentage of total variance in shape space [30], representing the percentage of variance related to the repetition of the digitization process. Random error was 4.9%, which was considered low and comparable to previous similar investigations [29,34].

## 3. Results

### 3.1. Sexual Dimorphism and Effect of Size (Allometry) in Craniofacial Shape

Craniofacial shape showed statistically significant sexual dimorphism in the present sample. This was assessed, separately for each configuration, with permutation tests (100,000 repetitions without replacement) on the mean Procrustes distance between males (*n* = 332) and females (*n* = 476). Despite the statistical significance, indicated by the *p*-values, the corresponding average Procrustes distances were small (Table 1). Thus, as also displayed in Figure 2, Figure 3, Figure 4 and Figure 5, the shape differences between males and females are barely noticeable upon visual observation of the respective structures.

Considering allometry testing, there was a significant effect of size on every shape configuration, in males and females. In females, centroid size explained 8,5% of the shape variation in the cranial base, 18.9% in the maxilla, 15% in the mandible, and 34.2% in the entire craniofacial configuration. In males, the respective effect sizes were 14.1% for the cranial base, 25.8% for the maxilla, 11.8% for the mandible, and 35.8% for the entire craniofacial configuration (*p* < 0.001 in all cases). After removing allometry, the shape differences between males and females, displayed in Appendix A, remain barely noticeable upon visual observation of the respective structures.

### 3.2. Number of Teeth and Craniofacial Shape

Age had a significant influence on the shape of all craniofacial configurations (0.120 ≤ η^2^ ≤ 0.439; *p* < 0.001). The number of missing teeth did not affect the shape of the cranial base in females or males (P_females_ = 0.110; P_males_ = 0.290); however, it had a strong effect on the shape of the maxilla, the mandible, and the entire craniofacial configuration. In females, the number of missing teeth predicted 5.7% of variation in maxillary shape (*p* = 0.002), 5.6% of variation in mandibular shape (*p* = 0.001) and 14.3% of variation in the shape of the entire craniofacial configuration (*p* < 0.001). In males, these effects were stronger, with the number of missing teeth predicting 14.2% (*p* < 0.001), 11.2% (*p* < 0.001), and 19.2% (*p* < 0.001) of shape variation in the maxilla, mandible, and the entire craniofacial configuration. These results represent the effect of missing teeth on craniofacial shape, when third molars were not taken into consideration, and are summarized in Table 2. Similar results were found when third molars were taken into consideration and are exhibited in Appendix A.

Due to the strong presence of allometry within sex groups, the standardized residuals of shape PCs, after removing the effect of allometry, were extracted as shape variables and the regressions of shape against missing teeth and age were repeated. The results regarding the effect of missing teeth on craniofacial shape, when third molars were not taken into consideration and after removing allometry are summarized in Table 3. The respective outcomes when third molars were taken into consideration are provided in Appendix A. In both cases, the results were similar to those reported on Table 2 and Appendix A, which included the allometric component. After removing allometry, age had no effect on any craniofacial shape (Table 3 and Appendix A), indicating that within sex groups the detected allometry was primarily due to age.

Shape differences in relation to tooth agenesis are presented in Figures showing Procrustes superimpositions of PC derived shape morphings representing cases with no or few missing teeth and cases with maximum number of missing teeth. Figure 6 exhibits these differences when third molars are not taken into consideration (with allometry), and Appendix A refers to the shape analyses performed when missing third molars are also accounted for (with allometry). As shown on the superimpositions, individuals with missing teeth presented with a more retrusive maxilla, a more pronounced mandible, and smaller Frankfurt Horizontal to mandibular plane angle, compared to controls. The effects are similar for both sexes but are more pronounced in males and are indicative of a more concave profile shape and a decreased facial height. The effects were similar when considering third molars and without. The respective outcomes after removing allometry and when third molars are not taken into consideration are presented in Figure 7, and when considering third molars, in Appendix A. The effects in both cases were comparable to those presented without removing allometry.

### 3.3. Local Versus Generalised Effects of Tooth Agenesis on Maxillomandibular Structures

Exploratory analyses of these effects were performed with and without accounting for the third molars and showed similar findings. It was evident that when tooth agenesis was located only in the mandible, the shape of both the maxilla and the mandible tended to be affected similarly. On the contrary, when tooth agenesis was located only in the maxilla, there was no detectable effect in the mandible (Appendix A). Thus, in case of tooth agenesis, the maxilla seems to be affected independently of the jaw in which the teeth are missing.

## 4. Discussion

Several studies have investigated the morphology of the craniofacial complex in individuals with tooth agenesis [16,35,36,37,38]. They present conflicting results, either due to intrinsic methodological limitations, such as the ones caused by the application of conventional cephalometrics, or due to actual heterogeneity among different samples [14,15,16,39]. If the used methods have severe limitations, performing accurate assessments becomes merely impossible, primarily when the actual differences are subtle. A recent study applying geometric morphometric methods attempted to overcome these limitations, but the relatively small sample size might have influenced its outcomes [9]. In the present study, we investigated the association of the number of permanent teeth to the shape of the craniofacial complex, by applying geometric morphometric methods in a large sample of modern humans (*n* = 808). In agreement with previous studies [9,16], we identified a link between tooth agenesis and reduced profile convexity as well as a smaller facial height. The effects were similar for males and females, increasing as the severity of the tooth agenesis increased, and remaining unaltered by the inclusion of third molars in the analysis, as well as after removal of the allometry effect. Furthermore, in case of tooth agenesis only in the maxilla, there was no detectable effect in the mandible, whereas the maxilla seemed to be always affected independently of the location of missing teeth.

In modern humans, congenital absence of one or more permanent teeth has a prevalence of 22.6%, when considering the third molars, and of 6.4% when not [1,40]. This high prevalence, along with findings showing a steady reduction of the number of teeth in a dentition during human evolution [41,42,43,44], raises the question whether tooth agenesis in modern humans is the result of a continuing evolutionary process rather than an isolated developmental disturbance. Previous studies have shown that modern humans with less teeth have also smaller faces [11,12]. The reduction of facial size is a documented trend during evolution and concurs with a reduction in tooth number and tooth size [41,43,45]. The association between tooth number and craniofacial shape described in the present study provides further support to the notion that a broader relationship between number of teeth and overall craniofacial development exists and might be the result of a mechanism developed during evolution [11,12].

In the present study, separate analyses were performed taking into consideration the absence of third molars as well, since the specific teeth are frequently missing, even in people with no agenesis of other teeth [20,40]. To confirm the findings, we performed all analyses with and without including third molars, since previous studies show conflicting outcomes regarding the association of third molar formation to craniofacial size [9,14,46]. Here, the findings were not modified noticeably by the effect of third molars.

Current evidence shows that age has a significant influence on the shape of the entire craniofacial configuration, as well as on the shape of individual craniofacial structures [47,48,49]. This was also consistently evident in our results, and for that reason, the effect of age was controlled in all the applied regressions models. Additionally, we repeated all analyses after removing allometry, and the results were similar, apart from the effect of age on craniofacial shape that was totally eliminated. This indicates that within sex groups, the allometry was primarily due to age differences between individuals; thus, it was ontogenetic allometry [49]. After controlling for age, sex, allometry, and absence of third molars, the number of missing teeth had a significant effect on the shape of the maxilla, the mandible, and the entire craniofacial shape. Nevertheless, there was no significant association between tooth agenesis and the shape of the cranial base. This supports previous findings [9] and is not surprising, since the midline cranial base is considered a conserved structure with limited variation among humans [11,49,50].

As observed on superimposed shape morphings, representing cases with no or few missing teeth and cases with large number of missing teeth, individuals with tooth agenesis presented a more brachyfacial skeletal morphology, a reduced maxillary length, a more protruded mandible, and retruded alveolar processes. These differences were more notable as the number of missing teeth increased. Studies using conventional cephalometric measurements reported similar findings, which are summarized in a recent systematic review [16]. Several previous studies identified various gene polymorphisms or mutations that relate to mandibular protrusion, which have also been implicated in tooth agenesis [51,52,53,54,55]. This implies a genetic link between the two phenotypes that could provide a plausible explanation for the present findings. Furthermore, a recent study reported that tooth agenesis is more frequent in Class II/2 and Class III malocclusion types [56], and another study found that dental anomalies, including tooth agenesis, were most prevalent in Class III skeletal patterns and in hypodivergent individuals [57]. In line with our study, the above findings suggest the existence of broad genetic mechanisms that lead to reduced number of teeth and at the same time to shorter faces with protruded mandibles.

Regarding the location of tooth agenesis and craniofacial shape, it was evident that tooth agenesis in the mandible had a similar effect on the shape of both the maxilla and the mandible. On the contrary, tooth agenesis in the maxilla only affected maxillary shape. There is scarce information regarding this finding. Two conventional cephalometric studies on maxillary lateral incisors agenesis indicated that the effects on craniofacial shape were limited to the maxilla [58,59]. A geometric morphometric study compared individuals with and without mandibular second premolar agenesis and concluded that the cross-sectional mandibular shape on the agenesis side is different [60]. Moreover, a cephalometric study showed that isolated tooth agenesis in the mandible is associated to a more retrusive upper lip [36]. This agrees with the skeletal maxillary changes that we observed, as well as with the claim that there is a broader relationship between tooth agenesis and overall craniofacial development. This claim, however, cannot be made for the mandibular shape based on the present findings because the mandible was not affected by isolated maxillary tooth agenesis. This supports the modular nature of the mandibular structure [61], but there is no further related evidence in the current literature. Thus, our findings on the topic remain to be confirmed by future studies.

In accordance with previous findings [9,49], mean shape differences between sexes were also evident in this study. Despite their small magnitude, it was decided to evaluate males and females separately. The same approach has been adopted, and similar differences have also been found previously [9]. Separate analyses are also justified due to sexual dimorphism in tooth agenesis patterns [62], which might have introduced an additional confounding factor, had both sexes been pooled. Furthermore, women are more often affected by tooth agenesis [1], implying sex differences in the biological mechanisms of tooth formation. The difference in genetic background between sexes might explain the greater variation and effect in craniofacial shape of males with missing teeth, as compared to females. The effect of tooth agenesis in male craniofacial morphology, for example, the decrease of facial convexity, resembles the natural differences observed between sexes. Females tend to have a less protrusive chin than males [30] and chin protrusion is largely associated with tooth agenesis.

The advantage of the present study is its methodology. The geometric morphometric methods for shape analysis [39], along with the robust sample and the applied statistical approach, allow the detection of even minor effects on the shape of the craniofacial configuration. To our knowledge, there is only one published study that investigated this topic using a geometric morphometric approach [9]. The study drew useful conclusions that were overall similar to ours, though based on a smaller sample population. This allowed only for mean comparison tests between predefined groups, some of which had questionable power. Here, we tested a larger sample, which enabled the use of regression methods on shape variables. This provided the opportunity to incorporate various contributing factors in the model and investigate our hypothesis thoroughly both in terms of presence and of magnitude of effects.

### Limitations

The present sample was collected from the pre-treatment records of orthodontic patients. Thus, it might not be representative of the general population. However, orthodontic treatment is very common in the places of sample collection, and thus, this population is not expected to largely deviate from the general population. Furthermore, both the agenesis and non-agenesis populations were retrieved from the same archives, adding to the homogeneity of the sample.

Moreover, our sample was limited to white Europeans. This was decided because these individuals were highly represented in the places of sample collection and because of racial differences in tooth agenesis patterns and in facial morphology [1,63]. Thus, the inclusion of a limited number of individuals with different origin might have confounded the outcomes. Further research is needed to confirm the outcomes in individuals of other than European ancestry.

Finally, due to the two-dimensional nature of our data, effects on the transversal dimensions could not be assessed. Three-dimensional images would be more informative but are difficult to collect in large numbers due to concerns regarding radiation exposure.

## 5. Conclusions

In the present study, we investigated the association between the number of permanent teeth and the shape of the craniofacial complex and revealed that tooth agenesis is related to a less convex or to a concave profile, as well as a shorter face. The effects were similar for males and females; they increased as the number of missing teeth increased, and did not change by the inclusion of third molars or of allometry in the analysis. Furthermore, in cases with tooth agenesis only in the maxilla, the mandible appeared normal. On the contrary, maxillary shape was affected in all cases of tooth agenesis regardless of the location of missing teeth.

The association of teeth number with craniofacial shape suggests a broader relationship between number of teeth and overall craniofacial development, which might indicate the presence of a common evolutionary mechanism controlling those phenotypic events.

## Figures and Tables

**Figure 1 biology-11-00544-f001:**
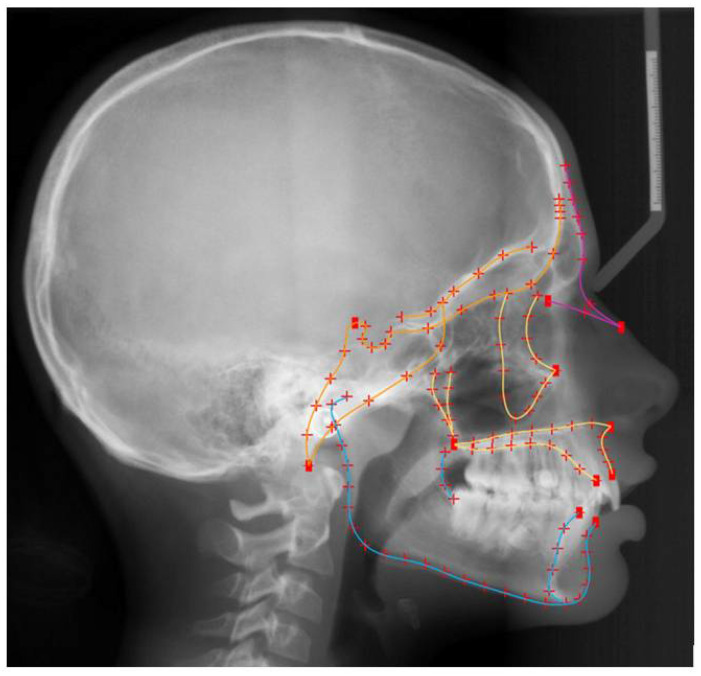
Material from Elias S. Oeschger et al. Number of teeth is associated with facial size in humans, Scientific Reports, published 2020, Springer Nature, licensed under CC BY 4.0. Craniofacial morphology was captured through the depicted landmarks. Digitization of the craniofacial complex (*n* = 808) with 15 curves, which included 116 semilandmarks (red crosses), and 11 fixed landmarks (red squares). Orange colour represents the structures of the cranial base, yellow the maxillary structures, blue the mandibular structures, and all lines together the entire configuration.

**Figure 2 biology-11-00544-f002:**
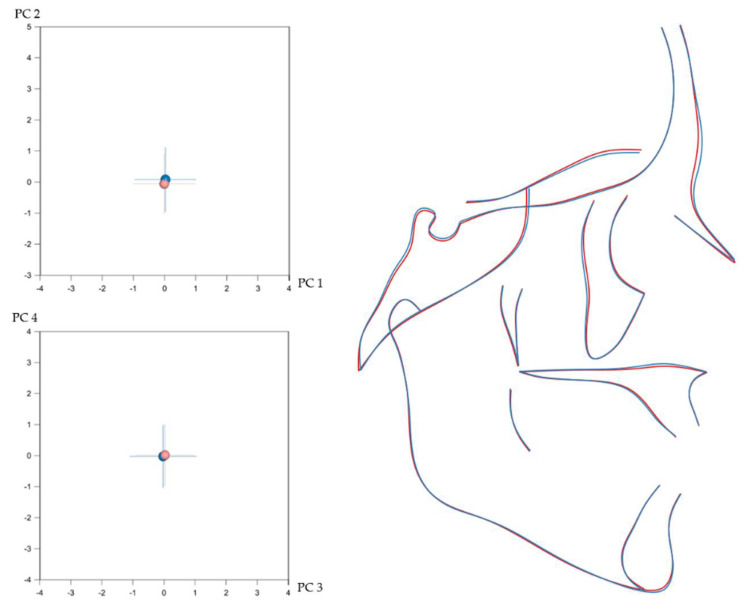
**Left**: Average difference between males (blue) and females (light red) in entire craniofacial configuration shape as explained by PC1 (21.8%)–PC2 (15.5%) (**top**) and PC3 (7.8%)–PC4 (6.2%) (**bottom**). The numbers in the parentheses represent the percentage of variation explained by each PC. (**Right**): Best fit superimposition of average male (blue) and average female (light red) craniofacial configurations.

**Figure 3 biology-11-00544-f003:**
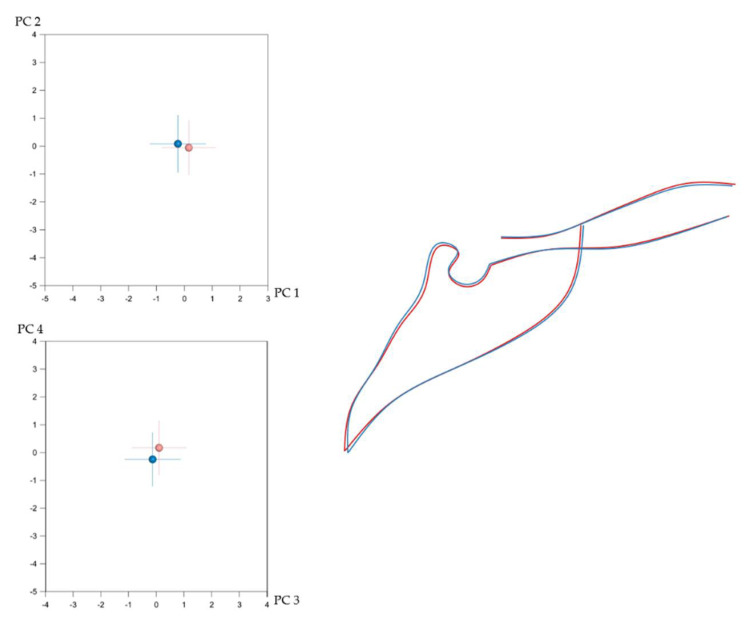
(**Left**): Average difference between males (blue) and females (light red) in cranial base shape as explained by PC1 (17.4%)–PC2 (16%) (**top**) and PC3 (11.2%)–PC4 (9.8%) (**bottom**). The numbers in the parentheses represent the percentage of variation explained by each PC. **Right**: Best fit superimposition of average male (blue) and average female (light red) cranial base configurations.

**Figure 4 biology-11-00544-f004:**
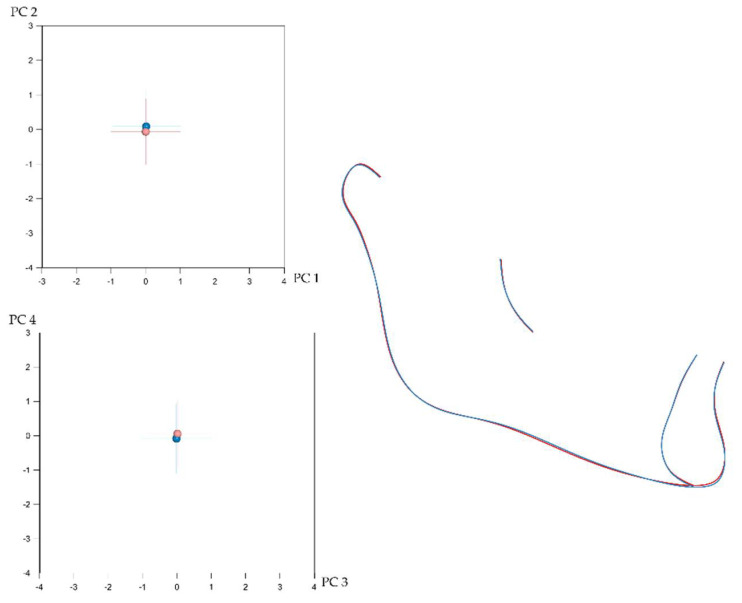
(**Left**): Average difference between males (blue) and females (light red) in mandibular shape as explained by PC1 (24.9%)–PC2 (21.4%) (**top**) and PC3 (13.9%)–PC4 (11.3%) (**bottom**). The numbers in the parentheses represent the percentage of variation explained by each PC. (**Right**): Best fit superimposition of average male (blue) and average female (light red) mandibular configurations.

**Figure 5 biology-11-00544-f005:**
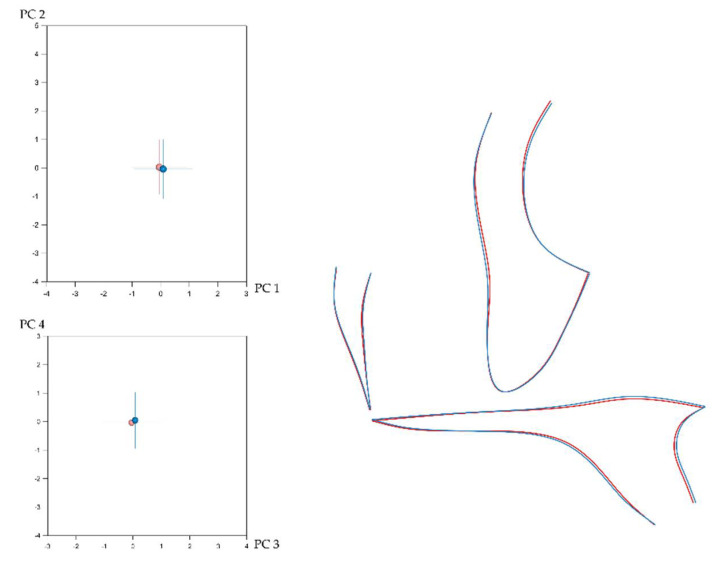
(**Left**): Average difference between males (blue) and females (light red) in maxillary shape as explained by PC1 (21.3%)–PC2 (15.6%) (**top**) and PC3 (11.1%)–PC4 (7.7%) (**bottom**). The numbers in the parentheses represent the percentage of variation explained by each PC. (**Right**): Best fit superimposition of average male (blue) and average female (light red) maxillary configurations.

**Figure 6 biology-11-00544-f006:**
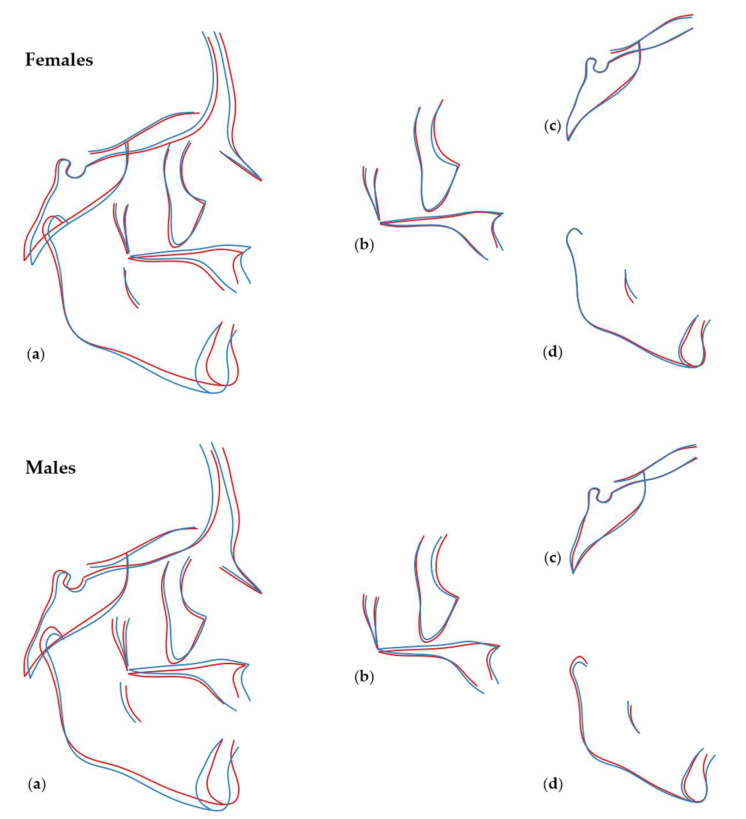
Regression of shape on number of missing teeth, not including third molars, in females (top row) and males (bottom row). The blue lines represent shape configurations in cases with no or few missing teeth (−4SD from average shape), and the red lines represent shape configurations in cases with large number of missing teeth (+4SD from average shape). The following shape configurations are displayed: (**a**) entire craniofacial configuration, (**b**) maxilla, (**c**) cranial base, (**d**) mandible. Cranial base differences were not statistically significant.

**Figure 7 biology-11-00544-f007:**
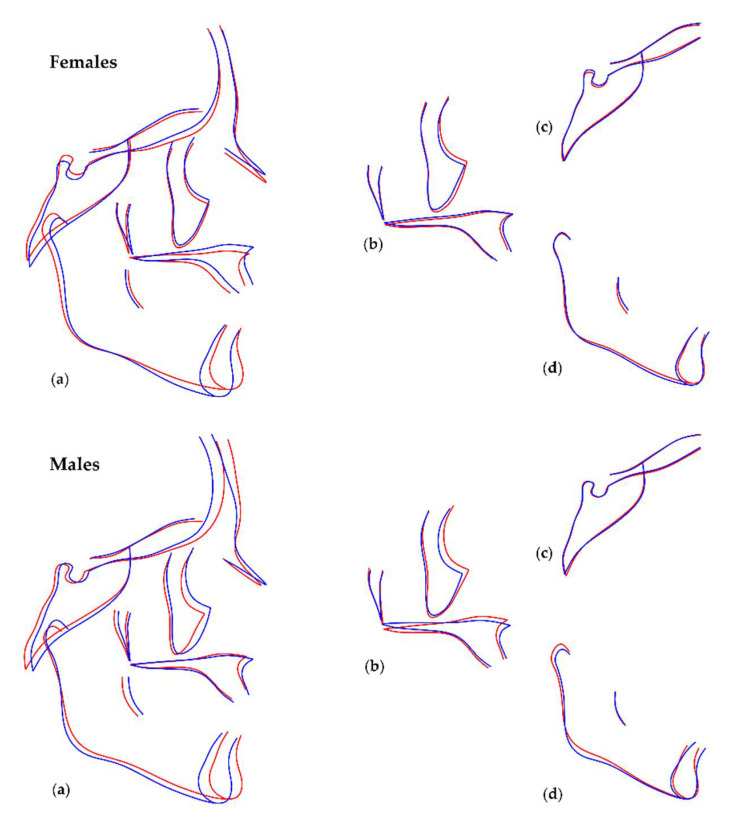
Regression of shape (after removal of allometry effect) on number of missing teeth, without considering third molars, in females (top row) and males (bottom row). The blue lines represent shape configurations in cases with no or minimum missing teeth (−4SD from average shape) and the red lines represent shape configurations in cases with maximum number of missing teeth (+4SD from average shape). The following shape configurations are displayed: (**a**) entire cranial shape, (**b**) maxilla, (**c**) cranial base, (**d**) mandible. Cranial base differences were not statistically significant.

**Table 1 biology-11-00544-t001:** Shape differences between male and female craniofacial configurations.

Shape	Mean Procrustes Distance	*p*-Value
Cranial base	0.01859	<0.001 *
Maxilla	0.01053	0.001 *
Mandible	0.00649	0.048 *^,1^
Entire craniofacial configuration	0.00954	<0.001 *

* *p* < 0.05, ^1^ Marginally significant result.

**Table 2 biology-11-00544-t002:** Multivariate regression of shape on age and number of missing teeth, without considering third molars, in female (*n* = 476) and male (*n* = 332) samples. Each shape configuration was described with the number of PCs explaining more than 85% of shape variation, as assessed with the broken-stick method.

Shape Configurations			η^2^	*p*-Value
Cranial Base (PC1–PC9)	Females	Age	0.120	<0.001 *
		Number of missing teeth	0.030	0.110
	Males	Age	0.122	<0.001 *
		Number of missing teeth	0.033	0.290
Maxilla (PC1–PC10)	Females	Age	0.142	<0.001 *
		Number of missing teeth	0.057	0.002 *
	Males	Age	0.199	<0.001 *
		Number of missing teeth	0.142	<0.001 *
Mandible (PC1–PC8)	Females	Age	0.159	<0.001 *
		Number of missing teeth	0.056	0.001 *
	Males	Age	0.231	<0.001 *
		Number of missing teeth	0.112	<0.001 *
Entire craniofacial configuration (PC1–PC18)	Females	Age	0.369	<0.001 *
		Number of missing teeth	0.143	<0.001 *
	Males	Age	0.439	<0.001 *
		Number of missing teeth	0.192	<0.001 *

* *p* < 0.05.

**Table 3 biology-11-00544-t003:** Multivariate regression of shape (without allometry) on age and number of missing teeth, without considering third molars, in female (*n* = 476) and male (*n* = 332) samples. Each shape configuration was described with the number of PCs explaining more than 85% of shape variation, as assessed with the broken-stick method.

Shape Configurations			η^2^	*p*-Value
Cranial Base (PC1–PC9)	Females	Age	0.000	1.000
		Number of missing teeth	0.031	0.095
	Males	Age	0.000	1.000
		Number of missing teeth	0.034	0.270
Maxilla (PC1–PC10)	Females	Age	0.000	1.000
		Number of missing teeth	0.051	0.006 *
	Males	Age	0.001	1.000
		Number of missing teeth	0.134	<0.001 *
Mandible (PC1–PC8)	Females	Age	0.000	1.000
		Number of missing teeth	0.060	<0.001 *
	Males	Age	0.000	1.000
		Number of missing teeth	0.114	<0.001 *
Entire craniofacial configuration (PC1–PC18)	Females	Age	0.001	1.000
		Number of missing teeth	0.138	<0.001 *
	Males	Age	0.001	1.000
		Number of missing teeth	0.192	<0.001 *

* *p* < 0.05.

## Data Availability

All data are available in the main text or the extended data. The protocols and datasets generated and/or analysed during the current study are available from the corresponding author on reasonable request.

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
