# Peer review of "Number of Teeth Is Related to Craniofacial Morphology in Humans"

_biology, 2022, doi:10.3390/biology11040544_

Round 1

Reviewer 1 Report

Dear authors,

  1. further research is needed to confirm the outcomes in other than European ancestry
  2. the general population is more representative for analysis
  3. tooth agenesis : including third molars and disregarding third molars? do you have some observations about more accurate analyses of specific tooth agenesis ex. incisors, premolars on craniofacial shape?could you add some more information from previous studies?

Reviewer 2 Report

This study tests an interesting hypothesis that tooth agenesis is reflected in craniofacial shape in humans which was suggested before but on limited sample size. The study design is appropriate; the authors have collected a sufficient sample size and used methods of geometric morphometrics which could point out even subtle shape changes between analyzed groups.  The authors also stated the limitations of their study. Overall, this is well-written manuscript, clear and precise. The major problem is that the authors did not include allometry in their analysis, i.e. they did not check if the present craniofacial shape differences between the control and agenesis groups, as well as between sexes, are related to the differences in the size of explored structures. Allometry (the size-related shape changes of morphological trait) is an important concept in studies of evolution and development. Since authors have previously shown that tooth agenesis have an impact on the skull size, and in this study different age groups are included, allometry must be included in present analyses of shape changes.

The specific suggestions and comments are given below.

Lines 40-41: The first sentences of the introduction are almost the same as the first sentences in the introduction of previously published papers. For example:

Oeschger et al 2020: “The congenital absence of one or more permanent teeth, known as tooth agenesis, is one of the most common dental anomalies in humans (prevalence: 6.4%)1 and is mainly attributed to genetic factors.”

Present MS: “The congenital absence of one or more permanent teeth, namely tooth agenesis, is one of the most common dental anomalies in humans. It has a prevalence of 6.4% and is shown to be more frequent in females than in males (1.22:1 females/males ratio).”

Please rephrase.

Lines 54-59: This part could be more detailed for a broader public. What are proposed relationships?  What are the major shortcomings of the cephalometric method?

Lines 125-129: In brackets, you say 404 patients with tooth agenesis, but in the further text you mention 493 patients with agenesis including third molars and 404 with agenesis without third molars included.  Please rephrase the first sentence to be clearer.

Lines 167-169: Figure 1 – which structure is represented in purple?

Statistical analysis: There is a relationship between size and shape, i.e. shape of morphological traits changes with size, the phenomena known as allometry (e.g. Mitteroecker et al., 2013; Klingenberg, 2016). Allometry can explain the evolutionary mechanism of shape changes and it is one of the major concepts in evo-devo studies. Morphometric analyses can be done on allometric (not corrected for size) and non-allometric components (corrected for size).

Since you showed that cranial size differs between groups with and without agenesis in previous papers, in this study different age groups are included, and shape changes presented in this MS are related to decrease of facial height, size should be included in the present analysis of craniofacial shape. You should calculate CS for your data sets (I would recommend CS of landmark configuration (structures) included in this shape analysis, not overall cranial size). You can include CS in GLM to check if there is an impact of size on craniofacial shape.  Another possibility is to do a permutation test to test if the allometry is significant.  To explore shape differences that are not related to size (overall, sexual dimorphism, age), you should do PCA analysis on the non-allometric component which is represented by residuals of multivariate regression of shape coordinates on centroid size. If allometry is significant, these results, which describe the effects of tooth agenesis exclusively on craniofacial shape, should be presented in MS.

I’m not familiar with the software that you used, but if it doesn’t include the possibility to use residuals for further analysis, you could try MorphoJ (https://morphometrics.uk/MorphoJ_guide/frameset.htm?index.htm) or some of packages for geometric morphometric analysis in R as geomorph (https://cran.r-project.org/web/packages/geomorph/geomorph.pdf ).

Suggestion: Multiple regressions of shape coordinates on CS and age could be presented as a Figure.

Klingenberg, C.P., 2016. Size, shape, and form: concepts of allometry in geometric morphometrics. Development genes and evolution226(3), 113-137.

Mitteroecker, P., Gunz, P., Windhager, S. and Schaefer, K., 2013. A brief review of shape, form, and allometry in geometric morphometrics, with applications to human facial morphology. Hystrix, the Italian Journal of Mammalogy24(1), pp.59-66.

Line 219: Is Bonferonni correction included? Even if it is, the difference in shape for the mandibula is borderline (P = 0.048), I would not interpret it as a significant result.

Line 260-262: This part is for the Materials & Methods section.

Line 266-271: These shape changes could be size related.

Lines 317-328: This part of the discussion should be changed or shortened. It is just repeating what is previously stated.

Lines 338-340: Already mentioned before.

Lines 354-356: How was this tested? Please explain in more detail.

Reviewer 3 Report

The manuscript entitled: "Number of teeth is related to craniofacial shape in humans". Although there are many studies on the craniofacial morphology and the tooth agenesis, the authors were investigate the association between the number of permanent teeth and the morphology of the craniofacial by applying geometric morphometric methods in a large sample of humans. Their findings are interesting among them regarding the location of tooth agenesis and craniofacial morphology, it was evident, that tooth agenesis in the mandible had a similar effect on the shape of the maxilla and mandible. On the contrary, tooth agenesis in the maxilla only affected maxillary morphology.

Authors should avoid using the word shape to refer to the craniofacial "shape" and instead use the term morphology. They should also use maxilla and mandible and not the term "jaws" which is more appropriate for no-humans.

Round 2

Reviewer 2 Report

Congratulations to the authors who have done a great job. I recommend this article for publishing.